

# Technical Note: Approximate Bayesian Computation to improve long-return flood estimates using historical data

Adam Griffin[1], Luke Shaw[2], Elizabeth Stewart[1]

[1]Centre for Ecology \& Hydrology, Wallingford, Oxfordshire, OX10 8BB. UK.
[2]Department of Mathematical Sciences, University of Bath, Claverton Down, Bath, BA2 7AY. UK

*Correspondence to*: Adam Griffin (adagri@ceh.ac.uk)

**Abstract.** For the Generalised Logistic distribution as used in UK flood frequency analysis, one standard approach for parameter estimation is through maximum likelihood methods. However, there can be problems with convergence to final estimates in cases where the true parameter values are extreme. This paper applies Approximate Bayesian Computation (ABC),

a likelihood-free approach popularised in statistical genetics, which generates candidate parameters and compares data simulated from those candidates to the observed data. Candidates whose data have summary statistics (Partial Probability Weighted Moments, PPWM) sufficiently close to those of the observed data are accepted as draws from the posterior distribution.

 The ABC-PPWM approach is applied to new historical data points to estimate the flood frequency distribution for the River

Severn at the Welsh Bridge in Shrewsbury, UK to improve the estimates of magnitudes of flood events with return period longer than the length of systematic records. Level data are derived from historical sources, and discharge estimates are obtained using data from upstream discharge gauging stations. When used in the ABC-PPWM approach, the results are at least as effective as the maximum likelihood methods, showing similar point estimates, and similar levels of variance. The estimates for the shape parameter for the GLO show some discrepancies, but this is known to be the most challenging to estimate given

the availability of only censored historical data. Unlike maximum likelihood methods, for which the estimate may not be obtainable, the ABC-PPWM approach is always successful.

## 1 Introduction

The use of historical data within flood frequency estimation is a current source of interest (Archer et al., 2016). This is especially true for quantifying magnitudes of floods with return periods significantly longer than the systematic records

available. In the UK, this is partially motivated by the National Flood Resilience Review (H.M. Government, 2016) which emphasised the need for estimation of the magnitude of rarer flood events, citing a series of flood events in Cumbria occurring since 2010 which have caused high levels of damage to these areas. Historical events pre-dating systematic records can give insight into the most extreme events, allowing better inference of these more damaging floods. Application of historical flood data in the UK dates back to the Flood Studies Report (FSR) (NERC, 1975), and was updated in (Bayliss and Reed, 2001).



Since the FSR, Stedinger and Cohn (1986) looked into incorporating historical events as well as paleoflood evidence to improve estimates and reduce uncertainty, and Hosking and Wallis (1986) also looked at the benefits of incorporating such data. Macdonald and Sangster (2017) looked at data from across the UK dating back to 1750 to identify flood-rich periods using newspaper excerpts, old level readings and other accounts to estimate equivalent discharge levels across a wide area.

However that paper focused less on absolute magnitude, rather looking for periods where larger floods occurred more frequently. Current UK methods use L-moment estimates and pooling-groups with hydrological similarity to improve long return period flood estimates. They focus on the modified Generalised Logistic (GLO) distribution as described in the Flood Estimation Handbook (Robson and Reed, 1999) with probability density function:

$$f(x) = \begin{cases} \dfrac{1 + \left(\frac{k}{\alpha}(x - \xi)\right)^{-\frac{1}{k+1}}}{\alpha\left(1 + \left(1 + \frac{k}{\alpha}(x - \xi)\right)^{\frac{1}{k}}\right)^2} & k \neq 0 \\[6mm] \dfrac{\exp\left[\frac{x-\xi}{\alpha}\right]}{\alpha\left(1 + \exp\left[\frac{x-\xi}{\alpha}\right]\right)^2} & k = 0 \end{cases}$$  (1)

for location parameter $\xi > 0$, scale parameter $\alpha > 0$ and shape parameter $k$. In the rest of this paper, we use $\theta = (\xi, \alpha, k)$ for brevity. In this paper, it is assumed that before the systematic record only the largest historical floods can be identified. Below some level, the *perception threshold* $X_0$, it is assumed that floods are not recorded. If no event in a year exceeded the perception threshold, then no AMAX value would be recorded.

In Section 2, the case study data are described, and the maximum likelihood method and the ABC method using PPWM are
15 outlined along with illustrative simulation examples. Section 3 shows the implementation of the ABC-PPWM method using the historical data from the Severn at the Welsh Bridge, and Section 4 discusses the findings.

## 2 Data and Current Methods

### 2.1 Maximum Likelihood

Maximum likelihood estimation (MLE) (Coles, 2001) is one of the key methods of parameter estimation for extreme value
distributions. Maximum likelihood estimation was the main tool in Wang (1990b) which incorporated censored historical data into flood frequency estimation. Stedinger and Cohn (1986) use a binomial term within the likelihood function to account for the probability of a number of historical events given a perception threshold. This method however, requires properties of continuity and boundedness of the likelihood function. In the case of the Generalised Logistic distribution, it was shown in Shao (2002) that for shape values less than 0, the MLE estimator may fail to converge using standard optimisation techniques
due to the unboundedness of the likelihood function. As the shape parameter of the true distribution decreases, the probability of failure to converge increases. This is not a problem which can just be ignored, since within the UK river network over 100 stations currently monitored by the NRFA are modelled using a GLO distribution with a shape parameter below 0. As such,



alternative methods should be implemented to ensure that in such cases, estimates can still be obtained which are comparable to the MLE method. For example, Parkes and Demeritt (2016) and Reis and Stedinger (2005) used a Bayesian approach with a Gibbs Markov Chain Monte Carlo sampler to incorporate historical events.

## 2.2 Approximate Bayesian Computation

Approximate Bayesian Computation (ABC) has seen use in situations where the likelihood function is either unable to be given analytically, or is slow/costly to evaluate or maximize but for which data are simple to simulate. It has been seen to be used to great effect in the areas of population genetics (Beaumont et al., 2002) and ecology (Csilléry et al., 2010). It is applied to flood frequency estimation for two primary reasons: firstly, it will be implemented to never fail to give parameter estimates for the Generalised Logistic distribution, and secondly to help with uncertainty quantification by generating posterior

distributions from which statistics of relevance can be computed. The method is outlined here, see Sunnåker et al. (2013) or Turner and Van Zandt (2012) for a more comprehensive introduction to the field, and Erhardt and Sisson (2015) for an application in extreme value analysis.

ABC aims to draw from the posterior distribution $p(\theta|X)$ of the parameters $\theta$ given the data $X$ by taking candidate parameter

choices, simulating data from them, and comparing summary statistics of this data to summary statistics of the actual observations. If they are close enough, the candidate parameters are kept, and are counted as an approximate draw from the posterior distribution. Otherwise they are discarded.

In our implementation we require: a sufficient summary statistic $S(x)$ for the GLO parameters, a distance metric $d(s_1, s_2)$ and a prior distribution $\pi(\theta)$ from which to draw candidate parameter choices.

Here a statistic $S = S(X)$ is *sufficient* for parameter $\theta$ if the value of the parameter conditional on the value of the statistic does not depend on the data $X$,

$$P[\theta = \theta_0 \mid S] = P[\theta = \theta_0 \mid S, X]. \tag{2}$$

In other words, the statistic explains everything one can know about the parameter; the data do not explain anything more.

The only known sufficient summary statistics for the GLO are the ordered AMAX record (which is of the same dimension as

the unordered AMAX record) and the MLE estimates of the GLO parameters. However, Erhardt and Sisson (2015) show that the L-moment estimators of the GLO parameters are also acceptable statistics for the ABC method. To account for the partial historical data in terms of censored data, Partial Probability Weighted Moments are applied (explained below).

The Mahalanobis distance metric (Mahalanobis, 1936) is used to determine similarity between summary statistics, scaling the different parameters according to their variability. In context, this means that a difference of $\delta$ in the location parameter is a

30 much smaller "distance" than a difference of $\delta$ in the shape parameter. More formally, given the summary statistics for two datasets $S_{(1)} = S(X^{(1)})$ , $S_{(2)} = S(X^{(2)})$ , their Mahalanobis distance is given by



$d_M(S_{(1)}, S_{(2)}) = \sqrt{(S_{(1)}^T \text{Cov}(S)^{-1} S_{(2)})}$. Here the covariance matrix $\text{Cov}(S)$ is estimated by simulating data using the parameter estimates from the observed data.

Theoretically, only those candidates whose summary statistic matches that of the observations are kept as exact draws from the posterior distribution (Erhardt and Sisson, 2015). However, in a continuous setting, the probability of equality is zero, and hence this is impractical to implement. Fortunately, if a sufficiently small but non-zero distance between summary statistics is permitted, the resultant estimate posterior distribution is a good approximation. In practice, a threshold $h$ is typically chosen such that a certain proportion of candidates are accepted. This threshold $h$ can be calibrated using a small trial run by taking a certain quantile of the distances recorded. In this work, 5% of candidates are accepted, and this value was used to decide $h$. Once $h$ is selected, candidates $X^*$ are kept if $d_M(S(X), S(X^*)) \leq h$, and rejected otherwise.

## 2.3 The Severn at the Welsh Bridge, Shrewsbury

The River Severn drains from the Welsh borders, and is one of the longest rivers in the UK (354 km) with the highest average discharge (61.2 cumecs) at the point at which it flows into the Severn Estuary and the Bristol Channel. Along its path is the town of Shrewsbury. The town has been subject to flooding on numerous occasions over the last 100 years (Black and Law, 2004) due to its location on the floodplain, and as such various flood alleviation schemes are in place within the town. With regard to flow monitoring, though no flow data are currently collected at the bridge, there is a stage level gauging station there which currently records 15-second observations.

Fifteen-minute stage level data are available from the Environment Agency for the last 15 years at the Welsh Bridge, but no flow data are recorded at this site. As a proxy for this, two nearby (less than 10 km distant) upstream flow gauging stations, Perry at Yeaton and the Severn at Montford are selected, both included in the National River Flow Archive's (NRFA) datasets (NRFA, 2018). Since no other major tributaries lie between these stations and the Welsh Bridge, an assumption was made that the flow was conserved, so flow at the Welsh Bridge was approximated as the sum of the flow from the two upstream stations. Annual maximum data for the two flow gauging stations (based on sub-daily observations obtained from the NRFA) were used with the Welsh Bridge level data, assuming that maximal levels corresponded to maximal flow values from the summed flow. These level and flow datasets cover an overlapping period of 30 years from which to derive a level-flow relationship: $Flow = 49.42(Level + 0.659)^{1.277}$, which fits very closely to the data up 500 cumecs as seen in Figure 1. This rating was used to convert historical level readings into approximate flow.




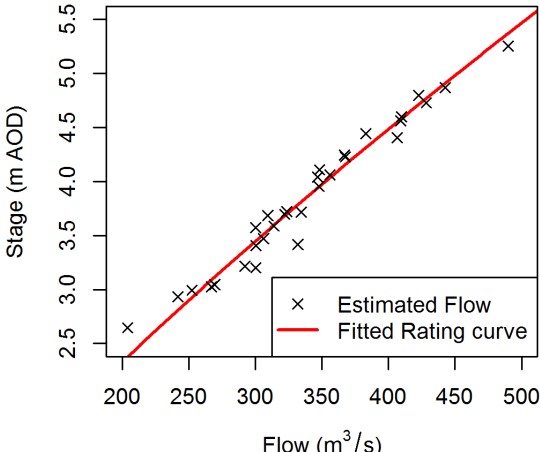

**Figure 1: Stage-Flow comparison of annual maxima series with standard NRFA rating curve fitted. RMSE = 11.83, less than 3% of median flow.**

## 2.4 Historical records

In order to obtain details of extreme events pre-dating systematic records, archived records were investigated. For this study, the Chronology of British Hydrological Events (CBHE, Black and Law (2004)) was used as an initial source to identify potentially valid extreme events, of which 73 were found. These were then specified using other sources such as local newspapers and records of the period. They were compared to the existing flow series for plausibility and for location of observation. Many such observations were made close to Shrewsbury Abbey, which although prominent is not close to the

river channel, and as such these observations were excluded. Note that this may bias the results due to excluding the largest floods. After this, 25 events were selected as appropriate and verified, and stage levels are reported in Table 1. It should be noted that the historical period is assumed to start at the earliest event. The derived rating curve for the Welsh Bridge was then used to obtain estimated flow values for the historical events, and these events are summarised in Fig. 2.

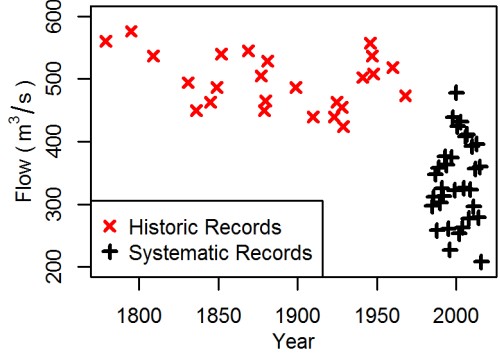

**Figure 2: Time series of derived systematic annual maxima with additional estimated flow for historical events back to 1800**



## 2.5 Partial Probability Weighted Moments

Partial Probability Weighted Moments (PPWMs) are an extension of the Probability Weighted Moments (PWM) used to determine L-moments as described in Hosking and Wallis (1997). These PPWMs, outlined in Wang (1990a) were introduced to incorporate historical flood information into estimating extreme-value distribution parameters.

Recall that the PWMs, $\beta_r$ are defined in Hosking and Wallis (1997) as

$$\beta_r = \int_0^1 F^{-1}(y)y^r \, dy \tag{3}$$

where $F^{-1}$ is the quantile (inverse) function of $F$. The standard PWM unbiased estimator $b_r$ is given by

$$b_r = \frac{1}{n} \sum_{i=1}^n x_i \frac{(i-1)(i-2)\cdots(i-r)}{(n-1)(n-2)\cdots(n-r)} \quad \text{for } r = 0,1,2,\dots,(n-1) \tag{4}$$

where the $x_i$ are sorted into ascending order.

To obtain the PPWM estimator, the standard PWM estimator is decomposed into two components either side of a preselected censoring threshold $X_0$. The lower bounded PPWM $\beta_r'$ and upper bounded PPWM $\beta_r''$ are given by

$$\beta_r' = \int_{F(X_0)}^1 F^{-1}(y)y^r dy \qquad\qquad \beta_r'' = \int_0^{F(X_0)} F^{-1}(y)y^r dy \tag{5}$$

The lower bounded estimator $b_r'$ is obtained by replacing $x_i$ in (4) by zero if $x_i \leq X_0$ and kept the same otherwise. Similarly, the upper bounded estimator $b_r''$ is obtained by replacing $x_i$ with zero precisely when $x_i > X_0$. The PWM estimator

is then the sum of the two components. In the case where there exists a systematic record along with a historical record, the censoring threshold $X_0$ is identified. The lower bounded estimate $b_r'$ makes use of all the data, both systematic and historical, but the upper bounded estimate $b_r''$ makes use of just the systematic record.

Parameter estimates are then obtained, as in Hosking and Wallis (1997), by computing estimates for the L-moment ratios which are given as linear combinations of the PPWMs. Here the mean (the first L-moment) is replaced by the median, as in

the Flood Estimation Handbook (Robson and Reed, 1999), due to the median's improved robustness to extreme events compared to the mean.

## 2.6 Algorithm Implementation

To implement the ABC algorithm, a prior distribution is required from which to draw candidate parameter vectors. There is no fixed way of determining this, but poor choices of prior distribution can lead to slow convergence to a posterior distribution

which is more representative of the underlying process. Typically, such priors are determined on a small subset of the data or based on expert opinion.

To achieve this, a trial run was performed with uniform priors around the L-moment parameter estimates based on the systematic data, with ranges wide enough that parameters on the boundaries of the uniform prior are never accepted. The results from the trial run were then used to approximate a Gaussian prior for the full run of the algorithm. This two-step process

was designed to be rigorous but efficient, as the first "cheap" step ensures viable parameter areas are not being ignored but computational time is not wasted on impossible parameter values.



## 3 Results

Here, the ABC-PPWM method are compared to the existing MLE method, and how the inclusion of historical data affects the flood frequency estimates. To begin with, a simulation study is performed, with synthetically created datasets drawn from a known GLO distribution, assuming a fully systematic record. Using true parameters $\theta = (40, 6, -0.2)$ an observation dataset

was created of 50 years of annual maxima. This is then used within the ABC-PPWM algorithm using a uniform prior about the L-moment estimates of the parameter values (data-driven) to generate 100000 candidates from which 3.3% were accepted, using an acceptance threshold obtained from a trial run. The posterior distributions of each of the parameters show reasonable values though the spread is quite high for the shape parameter (Figure 3, left).

Secondly, the systematic records are appended with a historical record. The length of the record is assumed known at 50 years

of systematic record, and a limited historical record over a period of 200 year (only values over 90th percentile threshold are kept). The right column of Fig. 3 shows that these also show fairly good convergence to a posterior normal distribution about the true values. However the shape parameter, due to the use of PPWMs, has a more noticeable bias due to the censoring of the data only retaining the most extreme values. An improved knowledge of the historical events, or a longer systematic record, would improve this estimate.

The Shrewsbury data are now used to consider the flood frequency curves estimates with and without new historical data sets under both the MLE method and the ABC-PPWM method. The flood frequency curves with and without the addition of the historical data points are computed (Figure 4).

For the MLE, 95% confidence intervals were obtained using the standard error. For the ABC-PPWM, 2.5% and 97.5% quantiles are used. Note that the ABC-PPWM method seems to underestimate when compared to the MLE estimate, but they

perform equally well in terms of variance of the posterior distribution. Once the historical data are included, the uncertainty is decreased but the posterior mean and the point estimates do not change by much, which agrees with previous work (Dixon et al., 2017). The ABC-PPWM posterior means stay inside the MLE method's 95\% bounds, and the parameter estimates are very similar (Table 2). However, it should be noted that the ABC-PPWM fit underestimates high flows. This is solely down to the shape parameter estimate, which if decreased by 0.1 from -0.03 to -0.13 would then overestimate high flows.

## 4 Discussion

As mentioned in Section 2.2 the rating equation does not take into account the flood going out of bank (i.e. past the top of the river channel). As such the rating continues to increase discharge nearly linearly, which leads to a highly underestimated discharge value. This can be seen on Fig. 4 where the biggest floods plateau substantially above a return period T of 50 years. Indeed no values of parameters would account for this extreme bend in the data, suggesting that the estimate could be improved

by a more comprehensive rating equation.





**Figure 3: Histograms illustrating posterior distributions of GLO parameters (location: top, scale: middle, shape: bottom) for simulated data. Left column shows only systematic data, right includes historical period. 95\% quantiles illustrated with true value and posterior mean.**

Computationally, the ABC-PPWM is more intensive for this simple example, due to the straightforward likelihood functions associated to this distribution. Where the MLE method took less than a second, the ABC-PPWM method took of the order of a minute to compute on the same computer. This would be improved if more informative priors were given, as determined by expert knowledge, or through more efficient simulation of AMAX systematic and historical series. The true strengths of the

10    ABC-PPWM method lie in its application to more complex models for which the likelihood methods cannot be applied, such as more design flood methods incorporating sedimentation and rainfall data, or flood frequency estimation incorporating more complex hydrological models, and it would be fruitful to try ABC in these cases in the future.




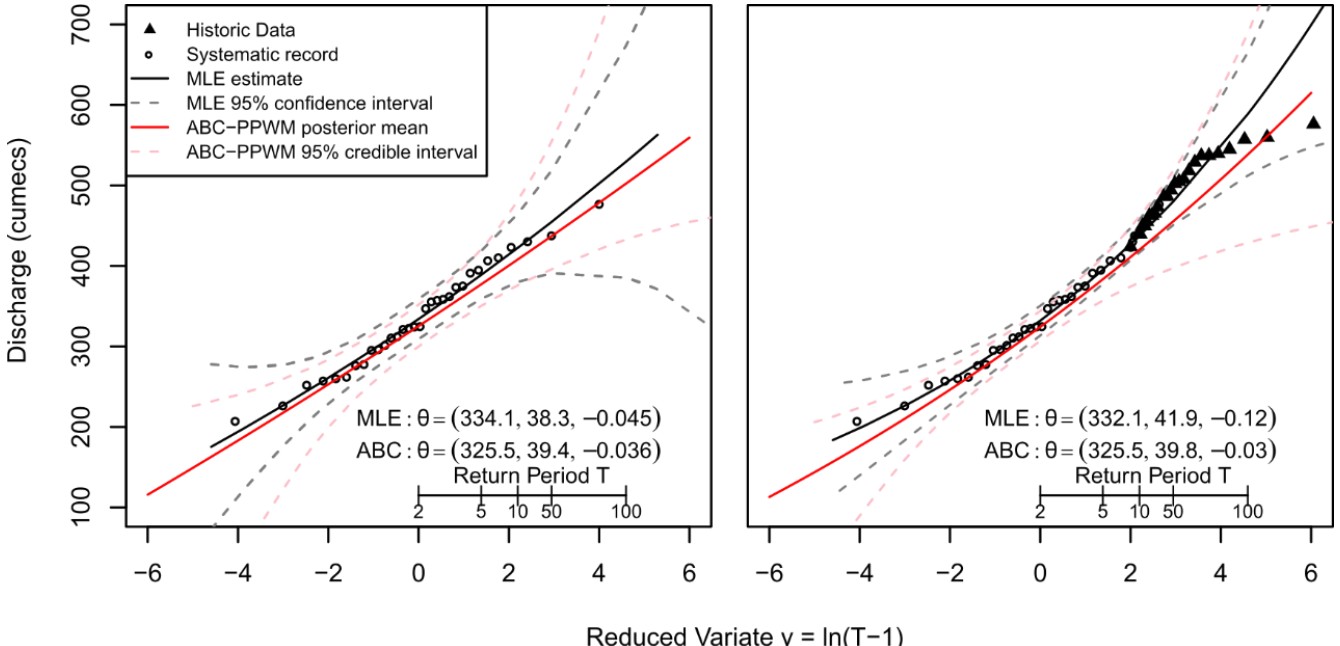

**Figure 4: Flood frequency curve using only systematic records (left) and incorporating historical record (right). $\theta$ corresponds to point ``mean'' estimates.**

## 5 Conclusions

An alternative method to the MLE approach for fitting a GLO distribution to historical and systematic data has been established, Approximate Bayesian Computation uses Partial Probability Weighted Moments, and although the method has a high computational cost, it has been shown to perform well, and is a viable consistent alternative in cases where MLE non-convergence is a problem. Historical data were collected at the town of Shrewsbury, containing information on past flooding of the Severn. These data were combined with systematic data from three gauging stations, two from the NRFA and one from

the Environment Agency, resulting in a data set of flood levels at Shrewsbury for a period of over 200 years.

Although the rating for the two gauge stations goes up to bankfull, some of the historical floods go beyond this, and as such, without extensive topographical modelling, it is not possible to improve the estimate for the flow-discharge rating equation beyond this point. It would be of interest in future work to perform inundation mapping to estimate the discharge associated to the largest floods, and hence the return periods. This was analysed with MLE and ABC-PPWM methods showing reduction

in uncertainty for higher return periods.

This method could also be applied to rainfall data, which is usually modelled using the Generalised Extreme Value (GEV) distribution (though the UK currently uses a six-parameter Gamma mixture model (Stewart et al., 2013)). The GEV and GLO distributions are closely related, and having a choice between which to use could broaden the analysis further. Similarly, the Generalised Pareto Distribution (GPa) is defined in terms of peaks-over-threshold (POT) data, and this paper artificially created



POT data by including the perception threshold. Hence an application of the ABC-PPWM method to POT data to fit a GPa distribution could be fruitful.

## Data Availability

AMAX series derived from 15-minute river stage data from the Environment Agency. Observations of river flow were provided by the National River Flow Archive (NRFA) at CEH Wallingford. Historical records were obtained directly from Shropshrire Archives, Shrewsbury, UK.

## Competing Interests

No competing interests are present in this work.

## Acknowledgements

This work in part undertaken as part of the NERC GW4+ Research Experience Programme. The authors would like to thank the Shropshire Archives and Matthew Weston at the Environment Agency for help concerning the data, and Alison Kay for offering useful advice.

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





| Year | Stage (m) | Source | Year | Stage (m) | Source | Year | Stage (m) | Source |
|------|-----------|--------|------|-----------|--------|------|-----------|--------|
| 1779 | 6.03 | E | 1877 | 5.51 | C | 1929 | 4.72 | C |
| 1795 | 6.18 | C | 1879 | 4.98 | C | 1941 | 5.49 | C |
| 1809 | 5.82 | E | 1880 | 5.13 | C | 1946 | 6.01 | A |
| 1831 | 5.41 | E | 1881 | 5.74 | C | 1947 | 5.82 | C |
| 1836 | 54.98 | E | 1869 | 5.89 | C | 1948 | 5.54 | C |
| 1845 | 5.11 | E | 1877 | 5.51 | C | 1960 | 5.64 | C |
| 1849 | 5.33 | C | 1879 | 4.98 | C | 1968 | 5.21 | A |
| 1852 | 5.84 | C | 1880 | 5.13 | C | | | |
| 1869 | 5.89 | C | 1881 | 5.74 | C | | | |

Table 1: The cleaned historical data collected for Shrewsbury. E = Eidowess's Journal (local newspaper), A = Shropshire Archives, C = Chronology of British Hydrological Events (Black, Law 2004)

| Method | Location $\xi$ | Scale $\alpha$ | Shape $k$ |
|--------|----------------|----------------|-----------|
| MLE | 332.1 (313.6, 350.5) | 41.9 (32.9, 51.0) | -0.12 (-0.27, 0.03) |
| ABC-PPWM | 325.5 (309.7, 341.3) | 39.8 (30.4, 50.0) | -0.03 (-0.20, 0.15) |

Table 2: Estimates for GLO parameters using MLE and ABC-PPWM posterior mean. 95\% confidence interval given in brackets for MLE, (2.5%, 97.5%) quantiles given for ABC-PPWM.