# Peer review of "Technical Note: Approximate Bayesian Computation to improve long-return flood estimates using historical data"

_Hydrology and Earth System Sciences, 2018_

## Referee Comment (RC1) · Anonymous Referee #1 · 2 Oct 2018

The paper is a valuable and interesting brief contribution to the literature on statistical techniques for incorporating irregular data in flood frequency estimation. It needs some more explanation and justification of assumptions, tidying up, minor revisions and a thorough proof read. I have not attempted to review all the mathematical details, or the description of the sufficient summary statistic.

Main comments 1. The discussion of post-FSR developments appears to end in 2001. It would be helpful to include more recent work or guidance, particularly since the Dixon et al paper is already referenced. 2. The reference to MacDonald and Sangster appears rather out of place and somewhat selective. 3. P2 line 9 mentions that

FEH methods improve long return period flood estimates: in the context, some readers might take this to mean that the FEH pooling method is a way of including historical data. 4. The discussion of current methods appears to be split between sections 1 and 2. Some re-structuring would help. 5. The introduction needs to explain why the research is needed, for example defining an objective, specifying a hypothesis to be tested or describing a gap to be filled. 6. It would help if the paper justified why the analysis is focusing on annual maxima. 7. The paper could usefully make clear how the annual maxima for the two upstream gauges were calculated, presumably by summing the two sub-daily flow series and then extracting the AMAX? 8. A significant assumption is made that the channel and floodplain hydraulics have not changed over the historical period, so that a current rating can be applied over the full period. The paper should explain how this assumption was checked: what investigations were carried out? For example, there is evidence that a weir was built in Shrewsbury in the early 20th century, and that earlier flood levels may have been elevated due to the presence of a mill wheel in one bridge arch. 9. The assumption that the historical period starts at the earliest event is weak. There are better ways to estimate the length of the period, for example as described in the Dixon et al (2017) reference already included. 10. The paper would be improved if it commented on whether the ABC method could be applied in the case where the magnitudes of the historical threshold exceedances are not known. 11. P.7 line 23: does the statement that high flows are underestimated rely on the assumption that the plotting positions of the observed floods are correct? 12. P. 7 line 26 refers to section 2.2 but in that section there does not appear to be a mention of how the rating treats out of bank flow. 13. P. 7 line 27: why test the method on a data series that is thought to be in error? 14. P. 7 line 28-30: might another explanation for the poor fit be that the chosen distribution is not an appropriate model, perhaps due to the effect of floodplain storage features (locally known as "argae") on the Severn upstream of Shrewsbury? 15. P. 8 line 3-4 is not entirely clear. Why can likelihood methods not be applied in some of the situations mentioned, such as using sedimentological data? Elsewhere the paper seems to imply that the true strength of

the ABC method is that it always converges.

Minor issues and presentation 1. The meanings of the phrases "new data points" and " point estimates" in the abstract are not clear. Similarly in the caption of Figure 4. Is "point" being used in some specialised sense here? 2. Some acroynms need spelling out in full when they first appear, such as GEV and AMAX. 3. There is repetition in the first few lines of page 3, and contradiction between 15 seconds and 15 minutes. 4. There are several uses of the phrase "as such" in a way that is grammatically questionable.

---

## Referee Comment (RC2) · Anonymous Referee #2 · 4 Oct 2018

The note proposes to use Approximate Bayesian Computations for flood frequency analysis and illustrates its application on the Severn river at Shrewsbury (UK). The idea may at first sight appear attractive but the content of the manuscript is questionable from several points of view:

1) It is first much too focused on flood frequency analysis practices in the UK. It should aim at being more general to be really useful for a large readership. The ABC method is for instance motivated by difficulties encountered when applying the classical maximum likelihood method for the calibration of generalized Logistic distributions (GLD). To my knowledge other types of distributions (Pearson III, generalized extreme value) are

preferred to the GLD distribution in many other countries. Are they affected by the same estimation problems ?

2) The main motivation for the use of the ABC method expressed on page 2 ("the maximum likelihood estimator may fail to converge for type I GLD) does not exactly correspond to the conclusions of the cited reference paper of Shao (2002). This paper explains the problem that can be encountered, but proposes also some simple solutions that can be easily implemented : include the so-called embedded distributions (Gumbel and 2-parameter reciprocal exponential distribution) in the tested solutions.

3) The implementation of the ABC method requires a measure of distance between the tested distribution defined by a set of parameters and the sample. This is mentioned, in a relatively confused way, on page 3 and 4, but the formula of this distance measure is never given. The reader understands from the abstract that this distance is based on probability weighted moments but that's all. The distance measure should be provided. Likewise, a justification for the 5% threshold should be provided somewhere, and maybe some kind of sensitivity analysis to this ad-hoc or arbitrary choice which determines the width of the "credibiliy intervals" which are computed (fig. 3 and 4).

4) The description of the ABC algorithm on top of page 4, recalls me the founding works of Hornberger and Spear (1980) on inference uncertainties. This method inspired K. Beven who developed later on the Glue method. Many researcher, and I am one of them, consider know that Bayesian inference and Monte Carlo Markov Chain algorithms introduced in hydrology 10 to 20 years ago, provide a rigorous and consistent method for inference problems, and especially for frequency analyses (see the abundant recent literature presenting implementations of Bayesian MCMC algorithms in flood frequency analysis). I have the feeling that the authors propose, under a knew name, the use of a 40-years outmoded method.

5) The cited reference list for flood frequency analysis is not up to date. The recent literature on Bayesian-MCMC flood frequency analysis is almost ignored by the note.

6) The reconstructed historical record series seems inconsistent with the recent measured series (figure 2). Most of the historic records lie beyond the adjusted rating curve for the present time (figure 1). The largest observed value over the recent 30 years period appears to have been exceeded 15 times over the past 170 years. There is a 6 percent chance for such an event to occur (1-15/170)^30. It is essential that historical records are thoroughly criticized before any implementation of statistical inference. Ideally, estimation uncertainties of historical flood discharges – that may be large, should be taken into account in the statistical inference and this is totally possible (see for instance Payrastre et al., WRR, 2011). The presented case study and implementation example is questionable and does not rely on the state of the art best practices. This is highly problematic since the note seeks to be exemplary.

7) The note is based on one single example which is probably not sufficient to demonstrate the usefulness, pertinence and robustness of the proposed method. The computed credibility intervals with two different methods appear close to one another on figure 4. But this could be the result of chance or of a calibration of the ABC threshold by the authors. It does therefore not demonstrate that the ABC credibility intervals do make sense: i.e. that they do accurately reflect the inference uncertainties. In fact, I highly suspect that they do not (see point 8). Moreover, is the proposed case study a problematic type I GLD case that would illustrate the benefit of the ABC approach? Apparently not since a MLE estimate is provided. The authors should illustrate that the non-convergence of MLE algorithms, which is the justification for the introduction of the ABC approach, is at least sometimes observed in real world.

8) The maximum likelihood confidence intervals in figure 4 are base on the so-called asymptotic Gaussian approximation which is clearly not suited (see the unrealistic decreasing lower bound on figure 4 left side). The authors should use the Bayesian MCMC algorithm, to computed credibility intervals (see for instance the BayesianMCMC command in the nsRFA package of the R software). The satisfactory outcome is the MLE estimates and the ABC posterior mean are close to one another. By the way,

the authors should explain why they selected the mean value rather than the parameter set and distribution that provided the best fit to the data, which would have been a more evident choice for the ABC algorithm. Is it because the results would have appeared less satisfactory?

9) The intervals computed with the ABC method are totally dependent on the selected threshold value. This dependence must be illustrated through a sensitivity analysis for instance as well as through the implementation of the method on various case studies. It must be acknowledged by the authors and it must be clear that no uncertainty level or probability can be attributed to these intervals. The figures and terms used (95% credibility interval is inadequate) are ambiguous and misleading.

———————————————————

---

## Author Comment (AC1) · 10 Dec 2018

The authors thank the reviewers and editors for their helpful contributions and comments. The full response to these comments (of both commenters 1 and 2) are contained within the zip file attached to this comment.

Please also note the supplement to this comment:
https://www.hydrol-earth-syst-sci-discuss.net/hess-2018-325/hess-2018-325-AC1-supplement.zip

325, 2018.